# Hiding in Plain Sight: Characterization of *Aeromonas* Species Isolated from a Recreational Estuary Reveals the Carriage and Putative Dissemination of Resistance Genes

**DOI:** 10.3390/antibiotics12010084

**Published:** 2023-01-04

**Authors:** Anna Luiza Bauer Canellas, Bruno Francesco Rodrigues de Oliveira, Marinella Silva Laport

**Affiliations:** 1Laboratório de Bacteriologia Molecular e Marinha, Instituto de Microbiologia Paulo de Góes, Universidade Federal do Rio de Janeiro (UFRJ), Av. Carlos Chagas Filho, 373, Cidade Universitária, Rio de Janeiro 21941-902, Brazil; 2Departamento de Microbiologia e Parasitologia, Instituto Biomédico, Universidade Federal Fluminense, Niterói 24210-130, Brazil

**Keywords:** *Aeromonas*, One Health, antimicrobial resistance, *bla*
_KPC_, virulence, horizontal gene transfer, WGS

## Abstract

Antimicrobial resistance (AMR) has become one of the greatest challenges worldwide, hampering the treatment of a plethora of infections. Indeed, the AMR crisis poses a threat to the achievement of the United Nations’ Sustainable Development Goals and, due to its multisectoral character, a holistic approach is needed to tackle this issue. Thus, the investigation of environments beyond the clinic is of utmost importance. Here, we investigated thirteen strains of antimicrobial-resistant *Aeromonas* isolated from an urban estuary in Brazil. Most strains carried at least one antimicrobial resistance gene and 11 carried at least one heavy metal resistance gene. Noteworthy, four (30.7%) strains carried the *bla*_KPC_ gene, coding for a carbapenemase. In particular, the whole-genome sequence of *Aeromonas hydrophila* strain 34SFC-3 was determined, revealing not only the presence of antimicrobial and heavy metal resistance genes but also a versatile virulome repertoire. Mobile genetic elements, including insertion sequences, transposons, integrative conjugative elements, and an IncQ1 plasmid were also detected. Considering the ubiquity of *Aeromonas* species, their genetic promiscuity, pathogenicity, and intrinsic features to endure environmental stress, our findings reinforce the concept that *A. hydrophila* truly is a “Jack of all trades’’ that should not be overlooked under the One Health perspective.

## 1. Introduction

Antimicrobial resistance (AMR) has quickly evolved into one of the biggest challenges in the global health sector, involving not merely the transfer of antimicrobial resistance genes (ARGs) between bacteria typically associated with humans, but also between bacteria widely distributed in animals and natural environments [1]. Hence, a multidisciplinary approach, such as One Health, is urgently needed to better comprehend the origins of ARGs and to monitor their spread. In particular, the growing acknowledgment of aquatic environments as “reservoirs” of ARGs indicates the importance of considering these habitats under the One Health perspective [2].

When considering natural environments, the genus *Aeromonas* stands out for its ubiquity and resilience. These bacteria can be found in a wide range of habitats and are able to thrive even under adverse conditions, such as in highly polluted waters [3]. The genus *Aeromonas* currently comprises 31 species [4], from which *Aeromonas caviae*, *Aeromonas dhakensis*, *Aeromonas veronii*, and *Aeromonas hydrophila* are most frequently described in cases of human infections [5]. In recent years, several studies have described potentially pathogenic and antimicrobial-resistant *Aeromonas* spp. in water matrices [6,7,8]. Of particular interest are the reports of such bacteria in recreational areas, where they come into close contact with humans, including immunocompromised individuals [9].

In this study, we investigated a total of 13 strains of *Aeromonas* spp. previously isolated on Thiosulfate Citrate Bile Salts Sucrose (TCBS) agar supplemented with ceftriaxone (2 µg/mL). These strains were isolated from Guanabara Bay (GB) [8], Rio de Janeiro, Brazil, and had their resistome features primarily investigated under a molecular and genomic perspective. GB is a tropical urban estuary located in the metropolitan area of Rio de Janeiro, and it is deeply impacted by anthropogenic pollution, especially by the disposal of untreated sewage [10]. Although often used as a recreational and fishing area, little is known about the risks involved with exposure to these waters, and studies of this nature could help to shed some light on the problem in GB and in other polluted areas. Since certain species of *Aeromonas* could be used as indicators of antimicrobial resistance in aquatic environments [11], our aim was to investigate strains of antimicrobial-resistant *Aeromonas* spp. from GB and characterize them regarding the presence of ARGs and heavy metal resistance genes (HMRGs). Apart from the presence of several resistance determinants, whole genome sequencing of a strain of *A. hydrophila* revealed key features that promote its survival in a deeply contaminated site, thus, expanding our understanding of the mechanisms employed by *Aeromonas* spp. to endure environmental stressors.

## 2. Results

### 2.1. Identification of Aeromonas Strains

Among the 13 antimicrobial-resistant strains of Aeromonas investigated, five (7SFC-1, 34SFC-21, 34SNFC-2, 34SFC-8, and 34FFC-2) were identified as Aeromonas jandaei (38.4%), three (34SFC-11, 34FFC-6, 34SFC-18) as Aeromonas veronii (23%), two (34SFC-4, 34SFC-9) as Aeromonas molluscorum, one (34SFC-2) as Aeromonas caviae, one (34FFC-3) as Aeromonas sanarellii, and one (34SFC-3) as Aeromonas hydrophila subsp. hydrophila. All strains were identified at the species level with 16S rRNA sequence identities higher than 98.75%, except for strain A. jandaei 34SFC-8, which showed an identity of 94.96%.

Phylogenetic analysis revealed a clustering of strains identified as *A. veronii*, *A. hydrophila*, and *A. molluscorum* with their corresponding reference sequence (*A. veronii* JCM 7375, *A. hydrophila* ATCC 7966, and *A. molluscorum* LMG 22214, respectively). The strains of *A. caviae* and *A. sanarelli* did not cluster with their reference sequence. Although somewhat distant from their reference sequence, strains of *A. jandaei* tended to cluster together (Figure 1).

### 2.2. Detection of intI1, Antimicrobial and Heavy Metal Resistance Genes

Antimicrobial-resistant strains were further characterized to assess the presence of class 1 integron-integrase, and antimicrobial and heavy metal resistance genes. Among these, six (46.1%) were *bla*_TEM_ positive. The second most frequently detected gene was *bla*_KPC_, identified in four (30.7%) strains. The *mcr-3* gene was detected in two (15.4%) and the *intI1* gene was detected in seven (53.8%) of the tested strains (Figure 2). On the other hand, *merA*, coding for mercury resistance, was the most often detected HMRG (10, 76.9%) and, out of a total of 11 strains positive for at least one HMRG, six (54.5%) strains also carried at least one ARG. The previously performed phenotypic antimicrobial susceptibility test results [8] are also shown in Figure 2. For the detection of heavy metal resistance, we relied on PCR assays, a more direct and eco-friendly approach.

### 2.3. Genomic Analyses

On account of its non-susceptibility to imipenem, the detection of ARGs and HMRGs, and the presence of the *intI1* gene, *A. hydrophila* strain 34SFC-3 was selected for whole genome sequencing. Genome-based identification confirmed the strain’s identification as *A. hydrophila,* and its complete genome consists of a single circular 4,961,387-bp chromosome with 60.9% GC content that encodes 4352 predicted coding sequences (CDSs). A total of 67 tRNA and five rRNA genes were predicted in the genome. Multilocus sequence typing assigned the strain to the ST378.

Upon investigation of the strain’s complete genome sequence, several ARGs were detected (Table 1), conferring resistance to ten different antimicrobial classes: beta-lactams, macrolides, sulfonamides, aminoglycosides, tetracyclines, colistin, fluoroquinolones, chloramphenicol, virginiamycin, and elfamycin. Noteworthy, *A. hydrophila* 34SFC-3 carries seven beta-lactamases, namely *bla*_TEM-1_, *bla*_CTX-M-2_, *bla*_KPC-2_, *bla*_OXA-726_, *imiH*, *bla*_OXA-18_, and *bla*_TOHO-1_. Interestingly, several mobile genetic elements were also detected, including an IncQ1 plasmid with 87.4% identity to the *Escherichia coli* plasmid RSF1010 (accession number: M28829.1). However, ARGs were not found in this plasmid, letting us infer that it may not play a role in the transfer of the detected genes in our survey. Five different insertion sequences (IS) were found, namely IS*Adh2*, IS*Ahy1*, IS*Aeme10*, and IS*As23*, which belong to the IS*1595* family, and IS*Psy43*, which belongs to the IS*66* family. For the detected ISs, different bacterial origins were designated, such as *A. dhakensis*, *A. hydrophila*, *Pseudomonas syringae*, *Aeromonas media*, and *Aeromonas salmonicida.*

Furthermore, we detected an *oriT* region, accompanied by a relaxase gene (*traI*) and two clusters of a type IV secretion system cluster in relative proximity (within a range of 300-kb upstream and 300-kb downstream from the *oriT* site) of six ARGs [*bla*_KPC_, *bla*_CTX-M-2_, *mphA*, *sul1*, *aac(6’)-Ib-cr6*, and *aac(3)-IId*]. The search for integrative conjugative elements (ICEs) revealed a putative ICE associated with a T4SS (31,665 bp) and a putative integrative and mobilizable element (IME; 9157 bp) (Figure 3). 

Along with ARGs, heavy metal resistance genes were found during the genomic analyses of strain 34SFC-3 (Table 2). Genes typically found in the *mer* operon, conferring resistance to mercury, were detected. In addition, other genes related to copper, silver, arsenic, cadmium, cobalt, zinc, and lead resistance were also identified.

Among the main virulence factors described for the genus *Aeromonas*, *A. hydrophila* strain 34SFC-3 harbors several genes encompassing different mechanisms, which are presented in Table 3.

### 2.4. Adaptation to a Polluted Environment

*A. hydrophila* 34SFC-3 harbors several features that could contribute to its survival in a deeply polluted environment. For instance, the genes involved in pyomelanin production, *phhA, phhB,* and *tyrR*, are present in its genome. The gene *hpd*, which codes for a 4-hydroxyphenylpyruvate dioxygenase and is essential for melanin biosynthesis, as well as the *hmgA* gene, coding for a homogentisate 1,2-dioxygenase, were also detected during genome annotation.

Sulfo-related reductase enzymes have also been identified in *A. hydrophila* 34SFC-3′s genome, which play a role in dealing with oxidative stress. For example, the strain harbors glutathione S-transferases (*gstB*), peptide methionine sulfoxide reductases (*msrA* and *msrB*), glutaredoxins (*grxC*), a zinc-type alcohol dehydrogenase-like protein, and 3-ketoacyl-CoA thiolases (*fadI* and *fadA*). Further, a thiopurine *S*-methyltransferase (*tpm;* involved in selenium cycling), a putative NAD(P)H nitroreductase YdjA (*ydjA;* associated with the reduction of nitroaromatic compounds) and an FMN-dependent NADH-azoreductase (*azoR*; associated with the reduction of azo groups) were detected. Enzymes involved in phosphonates degradation, such as 2-aminoethylphosphonate-pyruvate transaminase (*phnW*), and phosphonoacetaldehyde hydrolase (*phnX*) were also identified. 

## 3. Discussion

Under the One Health perspective, investigating antimicrobial resistance and bacterial virulence beyond the clinic is essential to better understand the mechanisms behind the origin and spread of clinically-relevant determinants. In the current study, we focused on strains previously classified as antimicrobial-resistant [8] and expanded their characterization to also encompass their heavy metal resistance profile and to determine the carriage of class 1 integron-integrases. The *intI1* gene plays a pivotal role in bacterial adaptation and antimicrobial resistance acquisition and dissemination [12]. More than 50% of the tested strains were positive for the detection of *intI1*, which has already been described in other environmental strains of *Aeromonas,* such as those isolated from zebrafish [13] and water and wastewater samples [14]. Also, the presence of the ARGs and HMRGs investigated in this study agrees with what is described in the genus *Aeromonas* [15,16,17,18].

All strains investigated were phenotypically resistant to at least two antimicrobials [8], raising the possibility that these strains are under selective pressure in the environment. Even though Guanabara Bay is a recreational and fishing area, it receives substantial input from untreated sewage, and agricultural and industrial effluents. Indeed, several studies have already reported the transport of ARGs, antimicrobial residues, heavy metals, and other pollutants in this kind of effluent [10,19]. Hence, the isolation of antimicrobial and heavy metal-resistant *Aeromonas* strains in a recreational environment raises attention to the potential risks associated with human exposure to these waters or the consumption of seafood from this region [20,21]. Although cases of *Aeromonas* spp. infections are not as frequent as other water-borne pathogens, they are considered emerging pathogens, and the arsenal of ARGs, virulence factors, and mobile genetic elements they carry may indeed pose a challenge to public health [5,22].

Regarding the antimicrobial resistance profile of the selected strains, it is of special interest to note the detection of the mobile colistin resistance gene *mcr-3* and the carbapenemase-encoding *bla*_KPC_ gene. The presence of *mcr-3* is well established in *Aeromonas* spp., and these bacteria may act as reservoirs of this gene in the environment [23,24]. The detection of *bla*_KPC_-harboring *Aeromonas* spp. has increased over the past years and they are usually described in hospital effluents and wastewater treatment plants [6,25,26]. Further genomic analyses confirmed not only the presence of the *bla*_KPC_ and *mcr-3* genes in strain 34SFC-3 but also unveiled several other ARGs, which could also pose a risk to public health. Some beta-lactamases are intrinsic to *Aeromonas* spp., such as *cphA*, which was detected in this study and confers carbapenem resistance [27]. Also, considering the arsenal of mobile genetic elements usually detected in this bacterial genus, *Aeromonas* spp. are possibly key disseminators of ARGs and likely mediate their transfer between typical clinical isolates and environmental bacteria [22]. For example, a strain of *A. caviae* isolated from a patient with pneumonia harbored IMEs associated with ARGs conferring resistance to different antimicrobial classes. Multidrug-resistant strains of *Aeromonas* isolated from zebrafish have also been shown to harbor class 1 integrons associated with ARGs, namely the *qacG-aadA6-qacG* array and *drfA1* [13]. Indeed, a highly mobilizable region was detected in the *A. hydrophila* 34SFC-3 genome, thus reinforcing its genetic promiscuity.

Apart from the antimicrobial susceptibility profile of the strains, the pathogenic potential was further investigated in *A. hydrophila* 34SFC-3. Several virulence-related genes were detected, mostly encompassing mechanisms of adhesion and secretion systems. The strain also harbors the *aerA* (aerolysin) and the *hlyA* (hemolysin) genes, suggesting its pathogenic potential. The occurrence of virulence genes in environmental strains is not unusual and they likely emerge as a consequence of exaptation, that is, factors primarily involved in environmental adaptation may also enable bacteria to become opportunistic pathogens, particularly in aquatic animals and, also, in humans [28].The strain was also assigned to ST378 which, according to the PubMLST database, had only been described once in China in 2014. Thus, its epidemiological relevance remains unclear.

With the exception of strain *A. jandaei* 7SFC-1, isolated from site 7, which is intermediately impacted by anthropogenic pollution in GB, all strains were isolated from site 34, considered one of the most polluted areas of GB [8,29]. In order to survive in environments with harsh conditions, bacteria must develop mechanisms to ensure their adaptation. For instance, nitroaromatic and azocompounds are often present in polluted environments. Nitroaromatic compounds usually originate from pharmaceutical, agrochemical, and explosives industries, while azocompounds are typically found in dyes used in textile industries [30]. Genomic analysis of *A. hydrophila* 34SFC-3 revealed the presence of a putative nitroreductase (*ydjA*) and an azoreductase (*azoR*), which have been previously investigated for their bioremediation potential. For example, a thermostable azoreductase from *Geobacillus stearothermophilus* exhibited broad substrate specificity and was considered a good candidate for whole-cell wastewater treatment [31]. Further, the several sulfo-related reductases here described have already been identified in strains of *Aeromonas* spp. isolated from effluents originating from a wastewater treatment plant, where they might offer protection against oxidative stress [6].

The detection of a thiopurine *S*-methyltransferase (*tpm*) also hints at the pollution levels in the waters where strain 34SFC-3 was isolated. This enzyme has already been described in the genus *Aeromonas* and is involved in the detoxification of metalloid-containing oxyanions and xenobiotics [32,33]. The 2-aminoethylphosphonate-pyruvate transaminase (*phnW*) and phosphonoacetaldehyde hydrolase (*phnX*) have already been reported in the *A. hydrophila* strain ATCC 7966^T^ and are involved in the degradation of phosphonates [34]. Another feature of some species of *Aeromonas* is the production of melanin and multiple enzymes previously reported in the synthesis pathways of this pigment were found. Melanin plays an important role in the protection against environmental stress, such as radiation, oxidative damage, and heavy metals, among others, and may also contribute to bacterial virulence [35]. For instance, a strain of *A. media* isolated from a lake in China was able to synthesize high levels of melanin, which was also proposed to be explored from a biotechnological perspective, namely in the photoprotection of bioinsecticides [36].

## 4. Materials and Methods

### 4.1. Water Sampling and Bacterial Strains

*Aeromonas* strains were isolated from three different sampling sites (1, 7, and 34; Appendix A) in Guanabara Bay (Rio de Janeiro, Brazil) in a previous report [8]. Briefly, filtered (membrane pore size 0.22 µm) and non-filtered 100 µL aliquots of water were spread onto TCBS agar supplemented with ceftriaxone (2 µg/mL). The cultures were incubated overnight at 25 °C and the colonies were isolated, purified, and cryopreserved as 20% glycerol stocks at −20 °C for further analyses. Thirteen strains were selected according to their phenotypic antimicrobial susceptibility [8].

### 4.2. Bacterial Identification and Phylogenetic Analysis

Bacterial identification was carried out by amplification and sequencing of the 16S rRNA gene. Briefly, genomic DNA was obtained using Chelex 100 resin [37]. Polymerase chain reaction (PCR) conditions were conducted as previously described [38] using the 16S rRNA universal primers, 27F (5′-GAGTTTGATCMTGGCTCAG-3′) and 1492R (5′-TACGGYTACCTTGTTACGACTT3′) [39]. PCR products were purified and directly sequenced using the 338F (5′-ACTCCTACGGGAGGCAGC-3′) at Biotecnologia, Pesquisa e Inovação (BPI, SP, Brazil). The resulting 16S rRNA sequences were quality-checked using Chromas 2.0 and taxonomic classification was performed with the EzBioCloud Database. An identity cut-off of 98.7% was considered for species delimitation [40]. All gene sequences were deposited in the NCBI GenBank database under the accession numbers: ON254898.1 (7SFC-1); ON259265.1 (34SFC-2); ON259266.1 (34SFC-3); ON259267.1 (34SFC-4); OP646809.1 (34SFC-8); ON259268.1 (34SFC-9); ON259269.1 (34SFC-11); ON259271.1 (34SFC-18); ON259272.1 (34SFC-21); ON259273.1 (34FFC-2); ON259274.1 (34FFC-3); OP646793.1 (34FFC-6); ON259275.1 (34SNFC-2).

Phylogenetic analysis of the thirteen strains was conducted by multiple sequence alignment with the Muscle algorithm in the MEGA software v.7. Reference sequences were downloaded from NCBI and a maximum likelihood (ML) tree was constructed using the Tamura-Nei model (Tn93+G+I) as the best-fit nucleotide substitution model with a bootstrap of 100 replications. *E. coli* NBRC 102203 (accession number: NR_114042.1) was used as an outgroup during the phylogeny inference.

### 4.3. Molecular Detection of Antimicrobial and Heavy Metal Resistance Genes 

PCR reactions were performed for the following genes encoding resistance to beta-lactams: *bla*_CTX-M-8_, *bla*_CTX-M-1,2_, *bla*_CTX-M-14_, *bla*_GES_, *bla*_TEM_, *bla*_SHV_, and *bla*_KPC_. Genes coding for mobile colistin resistance *mcr-1*, *mcr-2*, *mcr-3*, *mcr-4*, and *mcr-5*, as well as class 1 integron-integrase (*intI1*), were assessed in our survey. Heavy metal resistance genes *merA, merB, cusB, copA*, and *pbrA* were also investigated. Amplification reactions were carried out in a total volume of 25 µL using 10 ng of genomic DNA, 1x buffer GO TAQ Green Master Mix (Promega, Madison, WI, USA), and 20 pmol/L of each primer. All primer sequences and specific amplification conditions are detailed in Appendix A.

### 4.4. Whole Genome Sequencing, Assembly, and Annotation

Total DNA was extracted by the phenol-chloroform method [41]. Library preparation was performed using the Nextera DNA Flex Library Prep (Illumina) kit. DNA concentration and quantification were assessed using QuBit 4 with the DNA High Sensitivity test. Fragment size was visualized using Tapestation (Agilent Technologies) with the D1000 ScreenTape System. Genomic DNA was sequenced with the Illumina NextSeq 550. The quality of the read libraries was assessed using the FastQC v0.11.9 tool [42]. Pre-processing was performed with the Trim Galore v0.6.7 [43]. Genome assembly was conducted with SPAdes v. 3.15.4 [44]. Gene prediction was performed with Prokka v1.14.15 [45] and the genome completeness was calculated with the CheckM v1.1.11 tool [46]. The complete genome of strain 34SFC-3 is deposited at NCBI under the BioProject PRJNA894989.

### 4.5. Functional Genomic Analyses

The GTDB-Tool Kit 2.1.0 (GTDB-Tk) was used to taxonomically classify all strains based on the presence of 120 single-copy marker genes in their draft assemblies and the placement of their genomes in the Genome Taxonomy Database (GTDB) reference tree using the available database (GTDB version 1.4.5) [47]. Multilocus sequence typing of the whole genome sequence was performed with the web-based MLST sequence database [48] using the *Aeromonas* spp. MLST scheme (https://pubmlst.org/organisms/aeromonas-spp; accessed on 14 October 2022). Acquired antimicrobial resistance genes (ARGs) and/or chromosomal mutations were verified with the Resistance Gene Identifier tool (RGI 5.1.0) from the Comprehensive Antibiotic Resistance Database (CARD 3.0.7) [49], narrowing the criteria for “perfect and strict hits only” and the sequence quality for “high quality/coverage”. Additional ARGs were also searched manually. Virulence factors were investigated with the Virulence Factors Database (VFDB) [50]. Plasmids and insertion sequences (IS) were screened using PlasmidFinder [51,52] and ISFinder [53] (cut-off: 1 × 10^−5^), respectively. Integrative mobilizable elements (IMEs) and integrative conjugative elements (ICEs) were investigated with the ICEfinder web-based tool in ICEberg v.2.0 [54]. OriTfinder was used to detect the origins of transfer and three other transfer-associated modules, namely relaxase, type IV coupling proteins (T4CP), and type IV secretion system (T4SS) [55]. To predict integrons, the sequencing data were uploaded to the Galaxy web platform and the public server at https://galaxy.pasteur.fr/ (accessed on 24 October 2022) was used for the analysis to analyze the data with Integron Finder version 2.0.2 [56].

## 5. Conclusions

All in all, we demonstrated that *Aeromonas* strains isolated from a recreational estuary can harbor several ARGs, HMRGs, and virulence factors, suggesting that these bacteria should not be overlooked. Hence, our findings reinforce the importance of environmental assessment of the resistance and virulence determinants, thus providing valuable insights to tackle the emerging issue of antimicrobial resistance having *Aeromonas* as a paramount model in that framework.

## Figures and Tables

**Figure 1 antibiotics-12-00084-f001:**
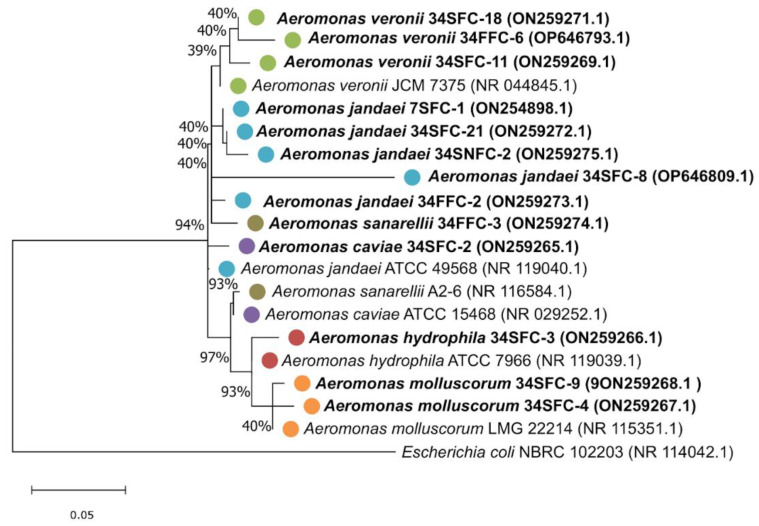
Maximum likelihood (ML) phylogenetic tree based on the 16S rRNA of *Aeromonas* spp. isolated from GB waters (in bold) and from reference sequences downloaded from NCBI. The colored dots represent different species, and the accession numbers are also depicted. Brown: *A. sanarellii*; purple: *A. caviae*; blue: *A. jandaei*; green: *A. veronii*; red: *A. hydrophila*, and orange *A. molluscorum*. *Escherichia coli* strain NRBC 102203 was used as an outgroup.

**Figure 2 antibiotics-12-00084-f002:**
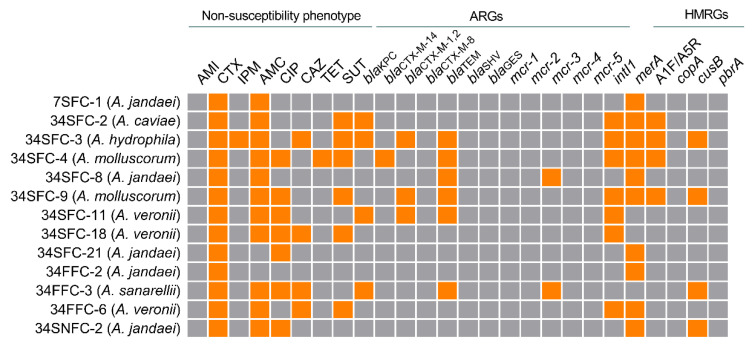
Antimicrobial susceptibility profile of the thirteen strains of antimicrobial-resistant *Aeromonas* from GB waters (Rio de Janeiro, Brazil), as well as the results of the detection of antimicrobial and heavy metal resistance genes (ARGs and HMRGs) and the integron-integrase gene (*intI1*). AMI: amikacin; CTX: cefotaxime; IPM: imipenem; AMC: amoxicillin-clavulanic acid; CIP: ciprofloxacin; CAZ: ceftazidime; TET: tetracycline; SUT: trimethoprim-sulfamethoxazole. Orange squares indicate positive PCR results, while gray squares indicate negative results.

**Figure 3 antibiotics-12-00084-f003:**
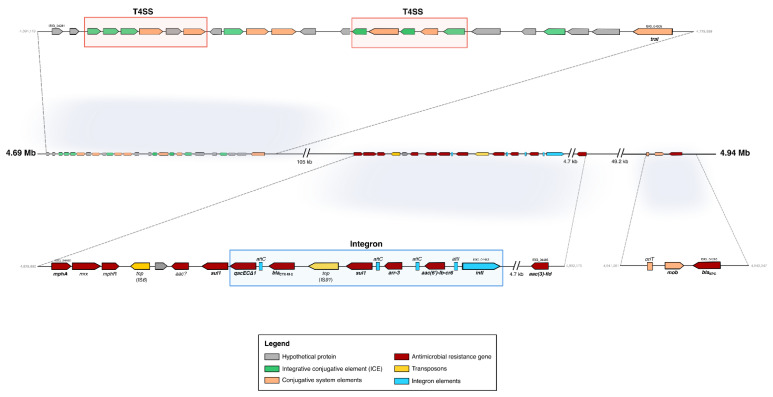
A highly mobilizable region in *A. hydrophila* 34SFC-3 genome. All genes are represented by arrows which indicate their genomic orientation. Genes encoding the respective mobile elements and ARGs are differently colored following the legend. Three key areas are zoomed in for greater comprehension of the genetic contexts surrounding ARGs and mobile elements.

**Table 1 antibiotics-12-00084-t001:** Antimicrobial resistance genes detected in the *A. hydrophila* 34SFC-3 genome, their corresponding gene family, and antimicrobials to which they confer resistance. The first part of the table refers to the results obtained in the CARD database, and the second, to the results obtained by manual curation.

Gene	Gene Family	Antimicrobials
TEM-1	TEM beta-lactamase	Monobactam, cephalosporin, penam, penem
*mphA*	Macrolide phosphotransferase (MPH)	Macrolide
*sul1*_1; *sul1*_2	Sulfonamide resistant *sul*	Sulfonamide
*qacE∆1*_1; *qacE∆1*_2	Major facilitator superfamily (MFS) antibiotic efflux pump	Disinfecting agents and antiseptics
CTX-M-2	CTX-M beta-lactamase	Cephalosporin
*arr-3*	Rifampin ADP-ribosyltransferase (Arr)	Rifamycin
*aac(3)-IId*	AAC (3)	Aminoglycoside
KPC-2	KPC beta-lactamase	Monobactam, carbapenem, cephalosporin, penam
*qacJ*	Small multidrug resistance (SMR) antibiotic efflux pump	Disinfecting agents and antiseptics
*adeF*	Resistance-nodulation-cell division (RND) antibiotic efflux pump	Fluoroquinolone antibiotic, tetracycline
OXA-726	OXA beta-lactamase	Carbapenem, cephalosporin, penam
*rsmA*	Resistance-nodulation-cell division (RND) antibiotic efflux pump	Fluoroquinolone antibiotic, diaminopyrimidine antibiotic, phenicol
*imiH*	*cphA* beta-lactamase	Carbapenem
*adeF*	Resistance-nodulation-cell division (RND) antibiotic efflux pump	Fluoroquinolone antibiotic, tetracycline
*aac(6’)-Ib-cr6*	AAC (6’)-Ib-cr	Fluoroquinolone antibiotic, aminoglycoside
*Escherichia coli* EF-Tu mutants conferring resistance to Pulvomycin	Elfamycin resistant EF-Tu	Elfamycin
*tetA*	Tetracycline resistance protein, class B	Tetracycline
*cat*	Chloramphenicol acetyltransferase	Chloramphenicol
*emrA*	Colistin resistance protein EmrA	Colistin
*emrB*	Colistin resistance protein EmrB	Colistin
*bla_3*	Beta-lactamase Toho-1	Penicillin G, ampicillin, oxacillin, carbenicillin, piperacillin, cephalothin, cefoxitin, Cefotaxime, ceftazidime, and aztreonam
*bla_1*	Beta-lactamase OXA-18	amoxicillin, ticarcillin, cephalothin, Ceftazidime, cefotaxime, and aztreonam
*yokD*	SPbeta prophage-derived aminoglycoside N (3’)-acetyltransferase-like protein YokD	Aminoglycoside
*vat*	Virginiamycin A acetyltransferase	Virginiamycin
*qacC*	Quaternary ammonium compound-resistance protein QacC	Quaternary ammonium compounds, antiseptics

**Table 2 antibiotics-12-00084-t002:** Heavy metal resistance genes detected in *A. hydrophila* 34SFC-3 genome. The table shows the heavy metal target, the corresponding heavy metal resistance genes detected, and their functional assignment, according to Prokka annotation.

Heavy Metal	Gene	Functional Assignment
Mercury	*merA, merA2*	Mercuric reductase
	*merC, merC2*	Mercuric transport protein MerC
	*merP, merP2*	Mercuric transport protein periplasmic component
	*merT, merT2*	Mercuric transport protein MerT
	*merR, merR1*	Mercuric resistance operon regulatory protein
Copper and silver resistance	*copA*	Putative copper-importing P-type ATPase A
	*copA2*	Copper-exporting P-type ATPase
	*copA3*	Copper resistance protein A
	*copA4*	Putative copper-importing P-type ATPase A
	*copB*	Copper resistance protein B
	*copR*	Transcriptional activator protein CopR
	*cusA1, cusA2*	Cation efflux system protein CusA
	*cusB*	Cation efflux system protein CusB
	*cusS*	Sensor histidine kinase CusS
	*cueO*	Blue copper oxidase CueO
	*cueR*	HTH-type transcriptional regulator CueR
Arsenic resistance	*arsA*	Arsenical pump-driving ATPase
	*arsC*	Arsenate reductase
	*arsD*	Arsenical resistance operon trans-acting repressor ArsD
	*acr3*	Arsenical-resistance protein Acr3
Molybdate resistance and homeostasis	*modA*	Molybdate-binding protein ModA
	*moaE*	Molybdopterin synthase catalytic subunit
	*moaD*	Molybdopterin synthase sulfur carrier subunit
	*moaA*	GTP 3’,8-cyclase
	*moaB*	Molybdenum cofactor biosynthesis protein B
	*moaC*	Cyclic pyranopterin monophosphate synthase
Heavy metal efflux pumps and transporters	*czcD*	Cadmium, cobalt and zinc/H(+)-K(+) antiporter
	*corA*	Magnesium transport protein CorA
	*zntR*	HTH-type transcriptional regulator ZntR
	*zntB*	Zinc transport protein ZntB
	*zntA*	Zinc/cadmium/lead-transporting P-type ATPase
	*fieF_1*	Ferrous-iron efflux pump FieF
	*cusB*	Cation efflux system protein CusB
	*acrA*	Multidrug efflux pump subunit AcrA

**Table 3 antibiotics-12-00084-t003:** Virulence genes detected in *A. hydrophila* 34SFC-3 genome, their corresponding functional assignment, and major virulence mechanism involved.

Gene	Functional Assignment	Mechanism
*flab1, flab2*	Flagellin B	Adhesion
*flgB*	Flagellar basal body rod protein FlgB	
*flgC*	Flagellar basal-body rod protein FlgC	
*flgD*	Basal-body rod modification protein FlgD	
*flgE*	Flagellar hook protein FlgE	
*flgF*	Flagellar basal-body rod protein FlgF	
*flgG*	Flagellar basal-body rod protein FlgG	
*flgH*	Flagellar L-ring protein	
*flgI*	Flagellar P-ring protein	
*flgJ*	Peptidoglycan hydrolase FlgJ	
*ecpD*	Fimbria adhesin EcpD	
*pile*	Fimbrial protein	
*pilQ*	Type IV pilus biogenesis and competence protein PilQ	
*ecpA*	Common pilus major fimbrillin subunit EcpA	
*pilT1, pilT2, pilT3*	Twitching mobility protein	
*tcpE*	Toxin coregulated pilus biosynthesis protein E	
*chew1,2*	Chemotaxis protein CheW	
*pomA1,2,3*	Chemotaxis protein PomA	
*tabA*	Toxin-antitoxin biofilm protein TabA	
*aerA*	Aerolysin	Hemolysins
*hlyA*	Hemolysin	
*hlyB*	Alpha-hemolysin translocation ATP-binding Protein HlyB	
*hlyD1, 2*	Hemolysin secretion protein D, chromosomal	
*vgrG1,2,3*	Actin cross-linking toxin VgrG1	
*apxIB1,2*	Toxin RTX-I translocation ATP-binding protein	
*bvg1, 2*	Virulence sensor protein BvgS	Virulence sensors
*phoQ*	Virulence sensor histidine kinase PhoQ	
*entE*	Enterobactin synthase component E	Siderophore
*entB*	Enterobactin synthase component B	
*entD*	Enterobactin synthase component D	
*Fur*	Ferric uptake regulation protein	
*epsC*	Type II secretion system protein C	Secretion systems
*epsE*	Type II secretion system protein E	
*epsF1,2*	Type II secretion system protein F	
*epsG*	Type II secretion system protein G	
*epsH*	Type II secretion system protein H	
*xcpW*	Type II secretion system protein J	
*epsL*	Type II secretion system protein L	
*epsM*	Type II secretion system protein M	
*outN*	Type II secretion system protein N	
*sctC*	Type 3 secretion system secretin	

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
