# Peer review of "Hiding in Plain Sight: Characterization of Aeromonas Species Isolated from a Recreational Estuary Reveals the Carriage and Putative Dissemination of Resistance Genes"

_antibiotics, 2023, doi:10.3390/antibiotics12010084_

Round 1
Reviewer 1 Report
This work is worthy publishing it as it has enough data obtained using current analytical techniques and it focuses to address health risk problem at the recreational center. However, since authors have not assessed the phenotypic antimicrobial and heavy metal resistance and relied on putative resistance only, they could recommend further work to be done on that angle as it will reveal the potential risks to individuals who will acquire the infection.
Author Response
Dear Reviewer 1,
Thank you very much for your time. The results of the antimicrobial resistance phenotype are shown in Figure 2 and the strains were selected according to their antimicrobial susceptibility, as formerly detailed in a previous study of our research group: Canellas, A.L.B.; da Costa, W.F.; Paranhos, R.; Laport, M.S. Diving into the unknown: Identification of antimicrobial resistance hotspots in a tropical urban estuary. Lett. Appl. Microbiol. 2021, 73(3), 270-279. doi: https://doi.org/10.1111/lam.13524
“For antimicrobial susceptibility testing, the following eight antimicrobials were used: amikacin, amoxicillin-clavulanic acid, cefotaxime, ceftazidime, ciprofloxacin, imipenem, tetracycline and trimethoprim-sulfamethoxazole, thus totalizing five different classes. The highest resistance percentages were detected for cefotaxime (56.6%), amoxicillin-clavulanic acid (55.0%), ciprofloxacin (26.6%) and trimethoprim-sulfamethoxazole (26.6%). Imipenem and tetracycline were the antimicrobials with the lowest resistance percentages (1.6 and 3.3% respectively). Interestingly, 50.0% of isolates showed concomitant nonsusceptibility to cefotaxime and amoxicillin-clavulanic acid, which could indicate the production of beta-lactamases.”
Phenotypic tests with heavy metal salts were not carried out to avoid the generation of toxic residues. Thus, only genotypic tests of resistance to heavy metals were performed in this study, in addition, to the inferences based on the genomic survey. In this revised version, we have taken particular care to make clear that these results were derived from molecular assays and the genome characterization of the A. hydrophila strain 34SFC-3.

Reviewer 2 Report
Aeromonads are not only ubiquitous environmental bacteria, able to rapidly colonize and cause opportunistic infections in humans and animals, they are also capable of promoting interactions and gene exchanges between the One Health components. In this study, 13 Aeromonas strains were isolated from a recreational estuary, and A. hydrophila strain 34SFC-3 was selected for whole genome sequencing. Based on gene and genome sequencing, several ARGs, HMRGs, and virulence factors were detected.
Specific comments:
1. Even for a communication instead of an article, thirteen was a small sample number to make the results convincing and meaningful.
2. For each isolate, antimicrobial susceptibility test according to CLSI was necessary. MDR bacteria were defined as bacteria that are non-susceptible to at least one antimicrobial agent in three or more antimicrobial classes not only based on drug resistance gene sequencing.
Please refer to Li, L., Yao, R., Heidemann, R., Zhang, Y., & Meng, H. (2022). Antibiotic resistance and polymyxin B resistance mechanism of Aeromonas spp . isolated from yellow catfish , hybrid snakeheads and associated water from intensive fish farms in Southern China. LWT, 166, 113802. https://doi.org/10.1016/j.lwt.2022.113802
Author Response
Dear reviewer 2,
Thank you very much for your time.
Aeromonads are not only ubiquitous environmental bacteria, able to rapidly colonize and cause opportunistic infections in humans and animals, they are also capable of promoting interactions and gene exchanges between the One Health components. In this study, 13 Aeromonas strains were isolated from a recreational estuary, and A. hydrophila strain 34SFC-3 was selected for whole genome sequencing. Based on gene and genome sequencing, several ARGs, HMRGs, and virulence factors were detected.
Specific comments:
- Even for a communication instead of an article, thirteen was a small sample number to make the results convincing and meaningful.
Reply: We agree that the manuscript should be a communication instead of an article. We would like to emphasize that Aeromonas samples of this manuscript were previously selected from a work published by the group (Canellas et al., 2022; doi:10.1111/lam.13524). Our main goal with this communication paper was not to provide an epidemiologically/statistically relevant sample size, but rather to characterize isolates that were previously isolated and get great insights based on molecular and genomic analyses. Indeed, the number of isolates during the isolation process was bigger (n = 344), but instead of working with all of those samples, we focused our time and resources on the most relevant ones in an One Health perspective, leading to the whole genome sequencing of one particularly interesting strain (34SFC-3) and generating information that raises attention to potential health risks.
- For each isolate, antimicrobial susceptibility test according to CLSI was necessary. MDR bacteria were defined as bacteria that are non-susceptible to at least one antimicrobial agent in three or more antimicrobial classes not only based on drug resistance gene sequencing.
Reply: The results of the antimicrobial resistance phenotype are shown in Figure 2 and the strains were selected according to their antimicrobial susceptibility, as formerly detailed in a previous study: Canellas, A.L.B.; da Costa, W.F.; Paranhos, R.; Laport, M.S. Diving into the unknown: Identification of antimicrobial resistance hotspots in a tropical urban estuary. Lett. Appl. Microbiol. 2021, 73(3), 270-279. doi: https://doi.org/10.1111/lam.13524
“Susceptibility to amikacin, amoxicillin-clavulanic acid, cefotaxime, ceftazidime, ciprofloxacin, imipenem, tetracycline and trimethoprim-sulfamethoxazole was assessed by disk diffusion method on Mueller-Hinton agar, according to CLSI’s guidelines. Escherichia coli ATCC 25922 was used as a quality control strain….Multidrug resistance was defined as the non-susceptibility to at least one agent in three or more antimicrobial classes, as described for other bacterial groups (Magiorakos et al. 2012)”*.
*Clin Microbiol Infect 2012;18(3):268-81. doi: 10.1111/j.1469-0691.2011.03570.x.
- Please refer to Li, L., Yao, R., Heidemann, R., Zhang, Y., & Meng, H. (2022). Antibiotic resistance and polymyxin B resistance mechanism of Aeromonas isolated from yellow catfish, hybrid snakeheads and associated water from intensive fish farms in Southern China. LWT, 166, 113802. https://doi.org/10.1016/j.lwt.2022.113802
Reply: Thanks for the indication. This reference has been cited and included in the manuscript's reference list in this newest manuscript version [24].
All indicated topics have been revised and improved.
Reviewer 3 Report
It is suggested to review the writing style of all the work in which the writing can be improved, considering the works in "past" and pointing out the importance of research on the subject, especially in the introduction and discussion sections, example: To summarize the findings of the NMA, MICD was used as defined in a review by Khattri et al. [27].
avoid if possible: Thirteen strains were selected according to 272 their antimicrobial susceptibility, as formerly detailed [8].
....conditions were conducted as detailed by Laport and colleagues [37]. - (correct is: et al.,)
It is important to review the order of the different sections of the work, since they are out of place, that is, (order): 1. Introduction, 2. Material and Methods, 3. Results, 4. Discussion, 5. Conclusions, and 6 References.
Review the style of the references, since some do not follow the same pattern that the journal suggests.
Author Response
Dear reviewer 3,
Thank you very much for your time.
- It is suggested to review the writing style of all the work in which the writing can be improved, considering the works in "past" and pointing out the importance of research on the subject, especially in the introduction and discussion sections, example: To summarize the findings of the NMA, MICD was used as defined in a review by Khattri et al. [27].
avoid if possible: Thirteen strains were selected according to 272 their antimicrobial susceptibility, as formerly detailed [8].
....conditions were conducted as detailed by Laport and colleagues [37]. - (correct is: et al.,).
Reply: The writing style has been revised in order to attend the reviewer’s remark.
- It is important to review the order of the different sections of the work, since they are out of place, that is, (order): 1. Introduction, 2. Material and Methods, 3. Results, 4. Discussion, 5.Conclusions, and 6 References.
- Review the style of the references, since some do not follow the same pattern that the journal suggests.
Reply: Following the instructions to the authors of the Journal Antibiotics (please, see https://www.mdpi.com/journal/antibiotics/instructions#preparation), the research manuscript sections should be organized such as follow: Introduction, Results, Discussion, Materials and Methods, Conclusions (optional). The references have been reviewed.
Round 2
Reviewer 2 Report
The concerns that we proposed are mostly properly addressed except for:
1. What is the threshold of the heat map (Fig 2) for determining positive, protein identity or other?
2. Whether drug-resistant genes are included in IncQ1 plasmid and whether they could be transferred?
Author Response
Author’s response - Round 2:
- What is the threshold of the heat map (Fig 2) for determining positive, protein identity or other?
Response: Figure 2 is not intended to be a heatmap. It is only a figure that describes the presence or absence of genes for each one of the tested Aeromonas strains, as described in the legend. This is made particularly clear in the figure caption, where red means absence and green indicates the presence of the targeted GRA.
- Whether drug-resistant genes are included in IncQ1 plasmid and whether they could be transferred?
Response: According to our genomic analysis, the detected IncQ1 plasmid does not contain any antimicrobial or heavy metal resistance genes and, therefore, would not play a role in their horizontal gene transfer for this particular Aeromonas strain. We added this information on item 2.3 for clarification in this new revised version of the manuscript.
Best regards
